Phylogenetic analysis of Fritillaria cirrhosa D. Don and its closely related species based on complete chloroplast genomes

Chen Qi 1
Wu Xiaobo 1
Zhang Dequan zhangdeq2008@126.com 1 2
1 College of Pharmacy and Chemistry, Dali University , Dali , Yunnan , China
2 Institute of Materia Medica, Dali University , Dali , Yunnan , China
Toonen Robert
Electronic publication date: 2019 Aug 21
Publication date: 2019
Volume: 7
Electronic Location ID: e7480
Received 2019 Apr 9; Accepted 2019 Jul 15
Copyright: ©2019 Chen et al.
Copyright year: 2019
Copyright holder: Chen et al.
License: This is an open access article distributed under the terms of the Creative Commons Attribution License, which permits unrestricted use, distribution, reproduction and adaptation in any medium and for any purpose provided that it is properly attributed. For attribution, the original author(s), title, publication source (PeerJ) and either DOI or URL of the article must be cited.
License URL: https://creativecommons.org/licenses/by/4.0/

Keywords: Fritillaria cirrhosa D. Don, Complete chloroplast genome, Closely related species, Taxonomically complex groups, Phylogenetic relationship

Funding: National Natural Science Foundation of China 31660081 Yunnan Provincial Science and Technology Department 2016FB144 This study was co-supported by the National Natural Science Foundation of China (31660081) and Yunnan Provincial Science and Technology Department (Grant No. 2016FB144).

==============================
Fritillaria cirrhosa D. Don, whose bulb is used in a well-known traditional Chinese medicine to relieve cough and eliminate phlegm, is one of the most important medicinal plants of Fritillaria L. The species is widely distributed among the alpine regions in southwestern China and possesses complex morphological variations in different distributions. A series of newly related species were reported, based on obscure morphological differences. As a result, F. cirrhosa and its closely related species constitute a taxonomically complex group. However, it is difficult to accurately identify these species and reveal their phylogenetic relationships using traditional taxonomy. Molecular markers and gene fragments have been adopted but they are not able to afford sufficient phylogenetic resolution in the genus. Here, we report the complete chloroplast genome sequences of F. cirrhosa and its closely related species using next generation sequencing (NGS) technology. Eight plastid genomes ranged from 151,058 bp to 152,064 bp in length and consisted of 115 genes. Gene content, gene order, GC content, and IR/SC boundary structures were highly similar among these genomes. SSRs and five large repeat sequences were identified and the total number of them ranged from 73 to 79 and 63 to 75, respectively. Six highly divergent regions were successfully identified that could be used as potential genetic markers of Fritillaria. Phylogenetic analyses revealed that eight Fritillaria species were clustered into three clades with strong supports and F. cirrhosa was closely related to F. przewalskii and F. sinica. Overall, this study indicated that the complete chloroplast genome sequence was an efficient tool for identifying species in taxonomically complex groups and exploring their phylogenetic relationships.

Introduction

Fritillaria L. is one of the most important genera in Liliaceae, which includes approximately 140 species of perennial herbaceous plants (Day et al., 2014; Tekşen, Aytaç & Pınar, 2010). Almost all of the species are distributed in the temperate regions of the northern hemisphere (Rix et al., 2001). There are 24 species in China and most of them possess important medicinal properties; these species include F. cirrhosa D. Don., F. ussuriensis Maxim., F. walujewii Regel., F. thunbergii Miq., and others (Chen & Helen, 2000; National Pharmacopoeia Committee, 2015). Of these species, F. cirrhosa is one of the major original plants of Fritillariae Cirrhosae bulbus, a famous traditional Chinese medicine, which is used to relieve cough and eliminate phlegm (National Pharmacopoeia Committee, 2015). It is mainly found in areas of high altitude in the southwest of China and grows in moist environments near bushes, meadows, and other similar habitats (Chen & Helen, 2000). However, this species exhibits complicated variations in morphology in different regions, especially in flower color and apex shape of bracts. Based on uncertain morphological differences, a series of newly related species were reported. As a result, F. cirrhosa and its closely related species constitute a taxonomically complex group that is difficult to be clearly distinguished based on morphological traits alone (Luo & Chen, 1996; Zhang, Zhang & Sun, 2001; Zhang & Cheng, 1998). Luo & Chen (1996) had proposed the concept of a “complex group of F. cirrhosa” which was composed of four species, namely F. cirrhosa, F. sichuanica, F. taipaiensis and F. yuzhongensis, based on uncertain morphological characteristics and geographical distributions. They also pointed out that F. sichuanica was possibly a hybrid among F. cirrhosa, F. przewalskii and F. unibracteata. This posits the question of whether the theory is reasonable and what would then be the phylogenetic relationships of the species in the so-called “complex group of F. cirrhosa” and their close relatives.

Over the past few decades, molecular methods have been widely used in plant evolution and phylogeny due to rapid development of molecular technologies. As a result, the well-known APG (Angiosperm Phylogeny Group) classification system was constructed based on the latest progress in plant molecular phylogenetics (Byng et al., 2016; The Angiosperm Phylogeny Group, 1998; The Angiosperm Phylogeny Group, 2003; The Angiosperm Phylogeny Group, 2009). Although molecular markers were also used for phylogenetic inference in complicated groups (Simmons et al., 2007), only a few species of Fritillaria adopted them to explore phylogeny (Çelebi et al., 2008; Wietsma et al., 2015). Nevertheless, gene fragments, especially nrITS and cpDNA genes, gained more attention due to the rapid development of DNA sequencing (Gao et al., 2010; Kress et al., 2005; Mishra et al., 2015). Day et al. (2014) elaborated on the evolutionary relationships of 92 Fritillaria species based on three plastid regions but most of the species were not well distinguished. Huang et al. (2018) used three plastid markers (matK, rbcL and rpl 16) and nuclear ITS to explore the phylogeny with 191 taxa in the tribe Lilieae (including 57 Fritillaria species) but the boundaries between a few species of Fritillaria remained ambiguous and needed further research. Meanwhile, the combination of nrITS and cpDNA genes was also adopted to reveal inter-specific relationships and to discriminate between the species of Fritillaria. Although these gene fragments had a preliminary resolution on certain species in Fritillaria, they could not be effective in discriminating between the closely related species (Khourang et al., 2014; Rønsted et al., 2005; Zhang et al., 2016). Overall, it is probably sufficient to use individual or combined regions based on Sanger sequencing in order to explore the phylogenetic relationships of major genera, but they are generally insufficient for complex groups or closely related species (Liu et al., 2017). Fortunately, with the emergence and development of next-generation sequencing (NGS), the complete chloroplast genome might be a better tool for discriminating between species and revealing the phylogenetic relationships of complex groups (Mardis, 2008; Parks, Cronn & Liston, 2009; Shendure & Ji, 2008; Tangphatsornruang et al., 2009).

In plants, chloroplasts (cp) are photosynthetic organelles providing the necessary energy for growth and are fundamental in the biosynthesis of starch, fatty acids, pigments, and amino acids (Gao et al., 2010; Neuhaus & Emes, 2000). Typically, angiosperm chloroplasts have a circular genome ranging from 72 to 217 kb and quadripartite structure composed of a large single copy region (LSC), a small single copy region (SSC), and a pair of inverted repeats (IRs) (Moore et al., 2010; Sugiura, 1992; Wang et al., 2015; Yurina & Odintsova, 1998). In contrast with nuclear and mitochondrial genomes, cp genomes are largely conserved in the gene content, organization, and structure. Moreover, they are typically inherited maternally in the angiosperm, which is beneficial in genetic engineering due to the lack of cross-recombination (Maliga, 2002; Tangphatsornruang et al., 2009). The initial cp genomes were sequenced from tobacco and liverwort (Ohyama et al., 1986; Shinozaki et al., 1986). The plastid genome exhibits an overwhelming superiority for use in species discrimination of complex taxa and has been widely used to reveal their unresolved phylogenetics (Bayly et al., 2013; Du et al., 2017; Henriquez et al., 2014), such as in Araceae, Arundinarieae, Lemnoideae, Myrtaceae, Nelumbonaceae, Amborella, Nymphaea, Citrus, Gossypium and Oncidium (Bayly et al., 2013; Carbonell-Caballero et al., 2015; Ding et al., 2017; Goremykin et al., 2003; Goremykin et al., 2004; Henriquez et al., 2014; Li et al., 2014a; Ma et al., 2014; Wu et al., 2010; Xue et al., 2012). In a recent study, Park et al. (2017) evaluated the relationships between F. ussuriensis and F. cirrhosa based on the chloroplast genome. Bi et al. (2018) explored the phylogenetic relationship of eight species representing each subgenus of Fritillaria using the complete chloroplast genome. Li et al. (2018) also adopted the plastid genomes to reveal inter-specific relationships among seven Fritillaria species that were mainly distributed in the Xinjiang province in China. Although these studies had partly revealed the classification and phylogenetics of Fritillaria and showed the power of a complete chloroplast genome, F. cirrhos a and its closely related species were not included and their relationships were still unresolved. Hence, we attempted to report the complete chloroplast genomes of F. cirrhosa and its related species, as well as to explore their phylogenetic relationships.

Here, we presented complete chloroplast genome sequences of several Fritillaria species using Illumina sequencing technology and performed comparative analyses of genomic information. Our aims were as follows: (1) to investigate the global structure patterns of eight plastid genomes in Fritillaria in this study; (2) to examine the variations of simple sequence repeats (SSRs) and other repeats (tandem, palindrome, forward, reverse, and complement repeats) among the eight Fritillaria plastid genomes; (3) to discover highly divergent regions that could be used as specific DNA barcodes for Fritillaria; and (4) to reveal phylogenetic relationships between F. cirrhosa and its closely related species. The study might provide better understanding of phylogenetic relationships of the complex group and afford sufficient genomic information to use in further research and the application of these medicinal species.

Materials & Methods

Material sampling

Eight species, including Fritillaria cirrhosa and its closed relatives were collected and used in this study (Fig. 1, Table S1). F. thunbergii was used as a supplemental outgroup for phylogenetic analysis. The related species were collected in the wilds of the Hengduan Mountains but F. thunbergii was cultivated in the Zhejiang province in China. Fresh, unblemished leaves were sampled from healthy, mature individuals and then dried with allochroic silicagel during the field work. Meanwhile, 3–5 individuals with flowers were collected and preserved as voucher specimens that were then used for morphological analysis and taxonomic identification. During the field work, geographic information, such as latitude, longitude, and altitude etc. was determined by Global Position System (GPS, Garmin) and morphologic traits (especially of the flower) were described immediately. All voucher specimens of Fritillaria were identified carefully by Dr. Dequan Zhang and deposited at the Herbarium of Medicinal Plants and Crude Drugs of the College of Pharmacy and Chemistry, Dali University.

Figure 1 Distribution of Fritillaria cirrhosa and its closely related species.

The distribution area of each species is drawn according to the records of Luo & Chen (1996), Liu, Wang & Chen (2009) and some existing voucher specimens (http://www.cvh.ac.cn/). Photos of representative living plants of eight Fritillaria species: (A) F. cirrhosa, (B) F. sichuanica, (C) F. taipaiensis, (D) F. yuzhongensis, (E) F. unibracteata, (F) F. przewalskii, (G) F. sinica, (H) F. dajinensis.Topographic data digital elevation modeling (DEM) data were required from the USGS website (https://glovis.usgs.gov/app?tour) with a 90-m spatial resolution grid.

DNA extraction, sequencing, and assembly

Total genomic DNA was extracted from about 100 mg of dried leaf material according to a modified CTAB method (Doyle, 1987; Yang, Li & Li, 2014). DNA quality was checked by electrophoresis on 1.2% agarose gel and then its concentration was determined using SmartSpecTM Plus Spectrophotometer (Bio-Rad, Hercules, CA, USA). DNA extracts were fragmented for 300 bp short-insert library construction and sequenced –2 × 150 bp paired-end (PE) reads on an Illumina HiSeq X-Ten instrument at Beijing Genomics Institute (BGI, Shenzhen, China).

The raw data was filtered using Trimmomatic v.0.32 (Bolger, Lohse & Usadel, 2014) with default settings. Then paired-end reads of the clean data were filtered and assembled into contigs using GetOrganelle.py (Jin et al., 2018) with Fritillaria cirrhosa (accession number: KF769143) as a reference (Li et al., 2014b), calling the bowtie2 v., blastN v. and SPAdes v.3.10 (Bankevich et al., 2012). The de novo assembly graphs were visualized and edited using Bandage Window dynamic v.8.0 (Wick et al., 2015) and then a whole or nearly whole circular chloroplast genome was generated.

Genome annotation and sequence submission

The plastid genomes were annotated by aligning to the complete chloroplast genome sequence published in GenBank (Fritillaria cirrhosa, accession number: KF769143) using MAFFT (Katoh & Standley, 2013) with default parameters, coupled with manual adjustment using Geneious v.10.1.3 (Kearse et al., 2012). The circular genome map was generated with OGDRAW v.1.2 (Lohse et al., 2013). Finally, the annotated chloroplast genomes of the nine Fritillaria species were submitted to GenBank (Table 1, Table S1).

Table 1 Summary of complete chloroplast genomes for eight Fritillariaspecies.

	Total (bp)	Large single copy (LSC,bp)	Small single copy (SSC,bp)	Inverted repeat (IR,bp)	GC%	Total
genes	Protein coding genes	tRNA	rRNA	Accession number in GenBank	
F. cirrhosa	151,998	81,755	17,545	26,349	36.9%	115	78	30	4	MH244906	
F. sichuanica	151,958	81,726	17,542	26,345	37.0%	115	78	30	4	MH244907	
F. przewalskii	151,983	81,744	17,539	26,350	36.9%	115	78	30	4	MH244908	
F. unibracteata	151,058	81,339	17,539	26,090	37.0%	115	78	30	4	MH244909	
F. taipaiensis	151,707	81,451	17,552	26,352	37.0%	115	78	30	4	MH244910	
F. yuzhongensis	151,645	81,417	17,526	26,351	37.0%	115	78	30	4	MH244911	
F. sinica	152,064	81,827	17,537	26,350	36.9%	115	78	30	4	MH244912	
F. dajinensis	151,991	81,723	17,540	26,364	36.9%	115	78	30	4	MH244913	

Genome comparative analysis

In this study, the multiple sequence alignment of chloroplast genome sequences was performed using MAFFT v.7.129 with default settings and adjusted manually in BioEdit v.7.0.9 (Hall, 1999; Katoh & Standley, 2013). The mVISTA software was used to compare the complete chloroplast genome of F. cirrhosa with eight other Fritillaria species, taking annotation of the chloroplast genome of F. cirrhosa (accession number: KF769143) as a reference. Default parameters were utilized to align the chloroplast genomes in Shuffle-LAGAN mode (Frazer et al., 2004). P-distance, GC content, and codon use were analyzed by the software MEGA v.7.0.26 (Kumar, Stecher & Tamura, 2016). DnaSP v.6.11 (Rozas et al., 2017) was adopted to calculate the variable and parsimony information sites and nucleotide diversity of five regions (whole chloroplast genome, large single copy, small single copy, inverted repeat regions, and protein coding genes). Additionally, the IR contraction/expansion regions were compared among the eight Fritillaria species.

Characterization of repeat sequences and SSRs

REPuter software was used to identify repeat sequences, including palindromic, complement, reverse, and forward repeats within the chloroplast genome. The following conditions for repeat identification were used in the analysis: (i) Hamming distance of 3, (ii) 90% or greater sequence identity, and (iii) a minimum repeat size of 30 bp (Kurtz et al., 2001). MISA was adopted to evaluate SSRs. The minimum thresholds were set to ten repeat units for mononucleotide SSRs, five repeat units for dinucleotide SSRs, four repeat units for trinucleotide, and three repeat units for tetranucleotide, pentanucleotide, hexanucleotide SSRs (Murat et al., 2011). In addition, tandem repeats in eight Fritillaria species chloroplast genomes were identified using Tandem Repeats Finder v.4.09 with the following settings: 80, 10, 50, and 500 for match probability, indel probability, minimum alignment score, and maximum period size, respectively (Benson, 1999).

Chloroplast genome analysis by sliding window

After using MAFFT v.7.129 to align the chloroplast genome sequences, BioEdit software was used to adjust the sequences manually (Hall, 1999; Katoh & Standley, 2013). A sliding window analysis was conducted for nucleotide variability (Pi) in the whole chloroplast genome using the DanSP. The step size was set to 200 bp, with a 600 bp window length (Rozas et al., 2017).

Phylogenetic analyses

The eight species of F. cirrhosa and its closely related species were used for phylogenetic analysis, to be supplemented with F. thunbergii (accession number: MH244914) and Lilium brownii F. E. Brown ex Miellez (accession number: NC_035588) as outgroups (Du et al., 2017). Furthermore, the available chloroplast genome sequence of F. unibracteata var. wabuensis (KF769142), which was a variety of F. unibracteata, was downloaded from GenBank for our phylogenetic analysis (Li et al., 2016). Phylogenetic inference was performed based on the following five data sets: (1) chloroplast genome sequence (only containing one IR), (2) large single copy region, (3) small single copy region, (4) inverted repeat region, and (5) protein-coding genes. The sequences were aligned using MAFFT and then edited by BioEdit manually (Hall, 1999; Katoh & Standley, 2013). Lengths of aligned sequences were shown in Table S8 . In order to explore the phylogenetic relationship of F. cirrhosa and its closely related species, Bayesian inference (BI), Maximum parsimony (MP) and Maximum likelihood (ML) methods were adopted for phylogenetic inference, respectively.

MEGA v.7.0.26 was used for MP analysis with 1,000 bootstrap replicates (Kumar, Stecher & Tamura, 2016). For BI and ML analysis, the best substitution models were tested based on Akaike information criterion (AIC) by jModelTest v.2.1.7 (Darriba et al., 2012). The best-fitting models in the analysis were GTR+I+G for LSC and SSC region, and GTR+I for others (Table S8). ML analysis was performed with RAxML v.8.2.4 (Stamatakis, 2014). And 1,000 replications were adopted to calculate the local bootstrap probability of each branch. BI analysis was conducted in MrBayes v.3.2.6 (Ronquist et al., 2012). The Markov Chain Monte Carlo (MCMC) algorithm was calculated for 1,000,000 generations with a sampling of trees every 1,000 generations. The first 25% of generations were discarded as burn-in. Stasis was considered to be reached when the average standard deviation of split frequencies was <0.01 and a consensus tree was constructed using the remaining trees.

Results

Chloroplast genome organization of Fritillaria chloroplast genomes

Nucleotide sequences of the eight Fritillaria chloroplast genomes ranged from 151,083 bp in F. unibracteata to 152,064 bp in F. sinica and shared the typical quadripartite structure, composed of a pair of IRs (26,090-26,364 bp) separated by the LSC (81,339-81,827 bp) and SSC (17,526-17,545 bp) regions (Table 1, Fig. 2). GC content of the complete chloroplast genomes was 36.9%-37.0% (Table 1). The content of the IR regions (42.5%) was higher than that of whole genome (36.9%), LSC (34.9%), and SSC (30.5%) in F. cirrhosa due to the presence of eight rRAN (55%) sequences in these regions (Table 2).

Figure 2 Gene map of Fritillaria chloroplast genomes.

Genes outside the circle are transcribed clockwise, and genes shown on the inside of the circle are counter-clockwise. Genes belonging to functional group are color-coded. The darker gray in the inner corresponds to GC content, and the lighter gray corresponds to AT content.

Table 2 Base composition in Fritillaria cirrhosa chloroplast genome.

	T/U%	C%	A%	G%	AT%	Length (bp)	
Genome	31.9	18.8	31.1	18.1	63.1	151,998	
LSC	33.3	17.9	31.9	17.0	65.1	81,755	
SSC	35.0	16.1	34.5	14.4	69.5	17,545	
IR	28.5	20.5	29	22.0	57.5	26,349	
tRNA	25.0	23.7	21.9	29.4	46.9	2,877	
rRNA	18.9	23.5	26.0	31.5	45.0	9,052	
Protein Coding genes	31.7	17.3	31.0	20.0	62.7	68,234	
1st position codon	24.6	18.1	30.9	26.4	55.5	22,745	
2nd position codon	32.2	19.9	29.9	18.1	62.0	22,745	
3rd position codon	38.3	14.0	32.1	15.6	70.4	22,744	

In the eight whole chloroplast genomes, a total of 115 genes were found, including 78 protein coding genes, 30 tRNA genes, four rRNA genes, and three pseudogenes (infA, ycf15 and ycf68) (Table 1, Fig. 2). The protein coding genes present in the chloroplast genome of eight Fritillaria genomes included nine genes for large ribosomal proteins (rpl2, rpl14, rpl16, rpl20, rpl22, rpl23, rpl32, rpl33, rpl36), 12 genes for small ribosomal proteins (rps2, rps3, rps4, rps7, rps8, rps11, rps12, rps14, rps15, rps16, rps18, rps19), five genes for photosystem I (psaA, psaB, psaC, psaI, psaJ), 15 genes for photosystem II (psbA, psbB, psbC, psbD, psbE, psbF, psbH, psbI, psbJ, psbK, psbL, psbM, psbN, psbT, psbZ), and six genes for ATP synthase (atpA, atpB, atpE, atpF, atpH, atpI) (Table 3, Fig. 2). Furthermore, 20 duplicated genes were found in the IR regions, as well as five protein coding genes, 11 tRNA genes and four rRNA genes. 26 protein coding genes possessed introns (Fig. 2).

Table 3 Gene contents in eight Fritillaria chloroplast genome.

Category for gene	Group of genes	Name of genes	
Self-replication	Large subunit of ribosome	rpl2I*, rpl14, rpl16*, rpl20, rpl22, rpl23I, rpl32, rpl33, rpl36	
	Small subunit of ribosome	rps2, rps3, rps4, rps7I, rps8, rps11, rps12I*, rps14, rps15, rps16*, rps18, rps19	
	DNA dependent RNA polymerase	rpoA, rpoB, rpoC1*, rpoC2	
	rRNA gene	rrn4.5I, rrn5I, rrn16I, rrn23I	
	tRNA gene	trnK-UUU*, trnI-GAUI*, trnA-UGCI*, trnG-GCC*, trnV-UAC*, trnL-UAA*, trnS-UGA, trnS-GCU, trnS-GGA, trnY-GUA, trnC-GCA, trnL-CAAI, trnL-UAG, trnH-GUGI, trnD-GUC, trnfM-CAU, trnW-CCA, trnP-UGG, trnI-CAUI, trnR-ACGI, trnI-CAUI, trnE-UUC, trnT-UGU, trnF-GAA, trnQ-UUG, trnR-UCU, trnT-GGU, trnM-CAU, trnV-GACI, trnN-GUUI, trnN-GUUI, trnV-GACI, trnG-UCC	
Gene for photosynthesis	Subunits of photosystem I	psaA, psaB, psaC, psaI, psaJ	
	Subunits of photosystem II	psbA, psbB, psbC, psbD, psbE, psbF, psbH, psbI, psbJ, psbK, psbL, psbM, psbN, psbT, psbZ	
	Subunits of NADH-dehydrogenase	ndhA*, ndhBI*, ndhC, ndhD, ndhE, ndhF, ndhG, ndhH, ndhI, ndhJ, ndhK	
	Subunits of cytochrome b/f complex	petA, petB*, petD*, petG, petL, petN	
	Subunit for ATP synthase	atpA, atpB, atpE, atpF*, atpH, atpI	
	Large subunit of rubisco	rbcL	
Other genes	Translational initiation factor	infA	
	Maturase	matK	
	Protease	clpP*	
	Envelope membrane protein	cemA	
	Subunit of Acetyl-carboxylase	accD	
	C-type cytochrome synthesis gene	ccsA	
	Open reading frames(ORF,ycf)	ycf1, ycf2I, ycf3*, ycf4, ycf15I, ycf68I	
Notes.

The I label after gene names reflect genes located in IR regions. Intron containing gene is indicated by one asterisk.

Protein coding genes, rRNA and tRNA were encoded by 44.89%, 5.96%, and 1.89% in the F. cirrhosa whole chloroplast genome, respectively, and the remaining 47.26% was non-coding regions. The 20 amino acids crucial for protein biosynthesis were encoded by 30 tRNA. Moreover, protein coding genes included 78 protein genes and the length was 68,234 bp, which comprised 22,396 codons (Table 2). Interestingly, among all of the encoded amino acids, leucine (10.32%) and cysteine (1.57%) were the maximum and minimum commonly detected amino acids, respectively (Table S2). Within the protein coding regions, the AT percentages for the first, second, and third codons were 55.5%, 62.0% and 70.4% in F. cirrhosa, respectively (Table 2).

SSR analysis of Fritillaria chloroplast genomes

Numerous SSR loci were found through the MISA analysis of nine Fritillaria chloroplast genome sequences. In total, five types of SSR (mononucleotide, dinucleotide, trinucleotide, tetranucleotide, and pentanucleotide repeats) were detected based on the comparison of eight Fritillaria cp genomes. A total of 78 perfect SSRs were found in F. cirrhosa (Fig. 3A). Similarly, 73, 74, 78, 79, 77, 76, and 75 SSRs were detected in F. sichuanica, F. przewalskii, F. unibracteata, F. taipaiensis, F. yuzhongensis, F. sinica, and F. dajinensis. Lengths of those SSRs ranged from 10 to 22 bp (Table S3). The most abundant type of SSR were mononucleotide repeats ranging from 51 bp in F. sichuanica to 56 bp in F. unibracteata, followed by dinucleotide repeats, tetranucleotide repeats, trinucleotide repeats, and pentanucleotide repeats (Fig. 3A). In the cp genome of F. cirrhosa, all mononucleotide repeats are composed of A (47.27%) and T (52.72%) motifs in the majority of dinucleotide SSRs are AT (64.29%) (Fig. 3B).

Figure 3 Analysis of simple sequence repeat (SSR) in eight Fritillaria cp genomes.

(A) Number different SSRs type detected in nine genomes; (B) frequency of SSR motifs in different repeat types of F. cirrhosa cp genome; (C) frequency of identified SSR in LSC, SSC, and IR regions; (D) frequency of identified SSR in IGS, CDS, and intron.

Further analysis revealed that most of the microsatellites were located in the LSC region, with a small portion distributed through the SSC and IR regions (Fig. 3C). Moreover, the SSRs in the genomes were distributed mainly in the intergenic spacer (IGS), with others dispersed at similar levels in introns and protein coding genes (CDS) (Fig. 3D). Seven protein coding genes in the SSR loci were rpoC2, cemA, ndhD, ndhG, ndhH, ycf1, and ycf2 in the CDS regions of the Fritillaria cp genome (Table S3).

Other repeats analysis of Fritillaria chloroplast genomes

A total of 63 repeats including tandem, palindrome, forward, reverse, and complement repeats were found in the F. cirrhosa chloroplast genome. Similarly, 65, 66, 70, 66, 73, 75, and 73 repeats were detected in F. sichuanica, F. przewalskii, F. unibracteata, F. taipaiensis, F. yuzhongensis, F. sinica, and F. dajinensis, respectively (Fig. 4A). Among these, tandem repeats, which had larger numbers than others, were mainly distributed in the intergenic spacer (IGS), with others dispersed in protein coding genes (CDS) and introns (Fig. 4B). The tandem repeats in the CDS regions were located in five protein coding genes (trnK-UUU, rps11, rps16, ycf1, and ycf2) of the plastid genomes (Table S4) and mainly ranged from 10 to 29 bp in length, whereas only one tandem repeat longer than 40 bp was found in the F. sichuanica genome (Fig. 4C). In the remaining four repeats, most occurred in the regions of the intergenic spacer, whereas some were found in the protein coding genes and intron (Table S5). Copy lengths with 30–44 bp were the most common. Moreover, the length of palindrome repeats more than 90 bp were found in four plastid genomes (F. sichuanica, F. przewalskii, F. yuzhongensis, and F. sinica). However, almost all of the lengths of the forward and reverse repeats were less than 59 bp in eight Fritillaria chloroplast genomes (Fig. 4D–4F, Tables S4–S6).

Figure 4 Analysis of large repeat sequences in eight Fritillaria cp genomes.

(A) Total of five repeat types; (B) frequency of tandem repeats in IGS, CDS, and intron; (C) frequency of tandem repeats by length; (D) frequency of palindromic repeats by length; (E) frequency of forward repeats by length; (F) frequency of reverse repeats by length.

Comparison of chloroplast genome among F. cirrhosa and related species

The annotation of F. cirrhosa (accession number: KF769143) was used as a reference for visualization analysis of the pairwise chloroplast genomic alignment between F. cirrhosa and its closely related species using mVISTA (Fig. 5). The alignment revealed a high sequence similarity across eight Fritillaria plastid genomes, which showed that the genomes were highly conserved. Furthermore, a vast majority of sequence variations were concentrated in the single copy regions, compared with the least number in the IR regions. This indicated that there were higher divergence levels in the single copy regions than that in the IR regions. Moreover, coding regions were less divergent than non-coding regions. Similarly, sequence divergence in the intron was higher than that in the exon. Highly divergent regions among eight Fritillaria chloroplast genomes were mainly located in the intergenic spacers, including atpH-atpI, rpoB-trnC-GCA, petN-psbM, psbM-trnD-GUC, trnT-GGU-psbD, trnS-GGA-rps4, trnT-UGU-trnL-UAA, accD-psaI, ycf4-cemA, and psbE-petL, but others (matK and ycf1) were distributed in protein coding regions.

Figure 5 Visualization alignment of nine Fritillaria cp genomes.

VISTA-based identify plot showing sequence identify among eight Fritillaria species using Fritillaria cirrhosa D. Don as a reference. The thick black line shows the inverted repeats (IRs) in the chloroplast genomes.

Expansion and contraction at the boundaries of IR regions of eight Fritillaria chloroplast genomes were revealed and a detailed comparison of four junctions of two IRs between F. cirrhosa and its closely related species was performed (Fig. 6). There were some differences in length compared with each region among the Fritillaria chloroplast genomes, but they exhibited striking similarities on the IR borders. Although IR regions were highly conserved, subtle structure variation was still observed in the chloroplast genomes. In contrast, ycf1 was mainly located in the SSC region ranging from 4,293 bp to 4,320 bp and others 1,230 bp in IRa region. The border between IRb/LSC extended into the rps19, but there were only 31 bp in the IRb region of F. cirrhosa. Moreover, variation was found in F. unibracteata, and ndhF was 24 bp away from the SSC/IRb border.

Figure 6 Comparison of LSC, SSC, and IR border regions among eight Fritillaria cp genomes.

Colored boxes for genes represent the gene position.

Evolutionary divergences and differences among the eight Fritillaria chloroplast genomes were compared using sequence distance and nucleotide substitutions. Across all the species, p-distance was 0.0003–0.0023, and the value of the nucleotide differences was 52-340 (Table 4). The p-distance in three Fritillaria (F. cirrhosa, F. przewalskii, and F. sinica) was between 0.0005–0.0008.

Table 4 Number of nucleotide substitutions and sequence distance in eight complete chloroplast genomes.

	F. cirrhosa	F. sichuanica	F. przewalskii	F. unibracteata	F. taipaiensis	F. yuzhongensis	F. sinica	F. dajinensis	
F. cirrhosa		311	112	314	335	310	117	311	
F. sichuanica	0.0021		328	95	290	261	331	52	
F. przewalskii	0.0007	0.0022		317	340	314	81	328	
F. unibracteata	0.0021	0.0006	0.0021		277	252	320	105	
F. taipaiensis	0.0022	0.0019	0.0023	0.0018		169	337	294	
F. yuzhongensis	0.0021	0.0017	0.0021	0.0017	0.0011		313	261	
F. sinica	0.0008	0.0022	0.0005	0.0021	0.0022	0.0021		333	
F. dajinensis	0.0021	0.0003	0.0022	0.0007	0.0020	0.0017	0.0022		
Notes.

The upper triangle shows number of nucleotide substitutions and the lower triangle indicates genetic distance in complete cp genomes among species.

Table 5 Variable site analysis in Fritillaria chloroplast genomes.

	Number of
sites	Number of
variable sites	Number of parsimony information sites	Nucleotide
diversity	
Complete cp genome	152,707	728	342	0.00172	
LSC	82,378	514	243	0.00223	
SSC	17,582	162	74	0.00332	
IR	26,372	27	13	0.00038	
Protein coding genes	68,709	237	112	0.00129	

Figure 7 Sliding window analysis of eight Fritillaria cp genomes (window length: 600 bp, step size: 200 bp).

X-axis: position of the midpoint of a window; Y-axis: nucleotide diversity of each window.

Divergence region in chloroplast genome of the F. cirrhosa and related species

Nucleotide diversity of highly variable regions was calculated with a sliding window (step size was set to 200 bp, with a 600 bp window length) to estimate the divergence level of different regions in the eight Fritillaria plastid genomes. Of these, the SSC region exhibited the highest divergence levels (0.00332) and IR regions had the least (0.00038) (Table 5). Furthermore, six regions with a relatively high variability, including 5 intergenic regions (trnS-GCU-trnR-UCU, rpoB-psbD, rps4-trnF-GAA, petA-psbL, and ndhF-ndhD) and one gene region (ycf1) from the genomes, were selected as potentially suitable gene fragments for the study of species identification and phylogenetics in Fritillaria (Fig. 7). All highly divergent sequences were found in the SC regions whereas no higher variable loci were found in the IR regions. The six highly variable regions included 257 variable sites which possessed 116 parsimony informative sites and their nucleotide diversity values ranged from 0.00455 to 0.00935 (Table S7). The petA-psbL showed the highest variability, the next more variable regions were rps4-trnF-GAA, ndhF-ndhD, ycf1 and rpoB-psbD, but that of trnS-GCU-trnR-UCU was the lowest.

Figure 8 Phylogenetic relationship of nine Fritillaria species inferred from Bayesian analyses (BI), maximum parsimony (MP), and maximum likelihood (ML) of different datasets.

(A) Chloroplast genome (Only contains one IR); (B) LSC region; (C) SSC region; (D) protein coding region. Number above nodes are support values with Bayesian posterior probabilities (PP) values on the left, MP bootstrap values in the middle, ML bootstrap values on the right.

Phylogenetic relationship of F. cirrhosa and related species

In this study, five datasets extracted from the eleven plastid genomes were used for phylogenetic analysis (Fritillaria thunbergii and Lilium brownie were used as outgroups). BI, MP, ML analyses were performed to construct phylogenetic trees using the datasets (Table S8) and topology structures of the previous three trees were nearly identical. Finally, the BI tree was adopted to present phylogenetic results, with the addition of support values from MP and ML analyses. The phylogenetic tree based on different datasets achieved higher support values, except the IR dataset (Fig. 8, Fig. S1). According to the trees, the eight species of Fritillaria were obviously divided into three clades (clade I, II and III). Clade I contained four species with strong support, namely F. sichuanica, F. dajinensis, F. unibracteata, and F. unibracteata var. wubuensis. It was revealed that F. sichuanica had a close relationship with F. dajinensis. Fritillaria taipaiensis, and F. yuzhongensis, both of which were distributed in the northern edge of the complex group and were clustered into one clade (II). The last clade (III) was composed of F. cirrhosa, F. przewalskii and F. sinica which revealed that F. cirrhosa was a sister species to the latter two species.

Discussion

Comparative analysis of Fritillaria chloroplast genomes

Eight plastid genomes in this study ranged from 151,009 bp to 152,064 bp, consisting of 115 genes with a GC content of 36.9%-37.0% (Table 1, Table 2 and Fig. 2). In the chloroplast genome of F. cirrhosa, the GC content of the IR regions (42.5%) was highest, which could be attributed to the presence of eight rRNA (55%) sequences in these regions (Table 2). The present results were similar to previous reports with a higher GC content in the IR regions (Bi et al., 2018; Li et al., 2018; Park et al., 2017). These studies might be beneficial for systematically recognizing the gene number, gene order, and chloroplast genome structure of Fritillaria. Furthermore, protein coding genes of the F. cirrhosa genome were encoded by 44.89% and the AT percentage of the third codon in them was 70.4%. Preference for a higher AT content at the third codon position has been also observed in other terrestrial plant chloroplast genomes (Asaf et al., 2016; Liu & Xue, 2005; Qian et al., 2013; Tangphatsornruang et al., 2009).

SSRs in the chloroplast genome (cpSSRs), which are 1-6 bp repeating sequences and distributed throughout the genome, have been used for the study of population genetics because of their high variability (Asaf et al., 2016; Pauwels et al., 2012; Powell et al., 1995). In this study, certain parameters were set as microsatellites of more than 10 bp are prone to slipped-strand mispairing (Raubeson et al., 2007; Rose & Falush, 1998). Five types of SSR (mononucleotide, dinucleotide, trinucleotide, tetranucleotide, and pentanucleotide repeats) were detected and the number of them ranged from 73 to 79 (Fig. 3). The detected SSRs were located in seven protein coding genes (rpoC2, cemA, ndhD, ndhG, ndhH, ycf1 and ycf2) of the Fritillaria plastid genomes. In the previous study, Bi et al. (2018) observed that five types of SSRs were located in nine protein coding genes (matK, rpoC1, rpoC2, cemA, ndhD, ndhG, ndhH, ycf1 and ycf2). Lu, Li & Qiu (2016) found that 15 SSRs were located in eight protein coding genes (rpoC2, cemA, rpl22, ndhD, ndhE, ndhH, ycf1 and ycf2) of three cardiocrinum plastid genomes. Therefore, all the studies strongly indicated that the chloroplast genome could be used for developing lineage-specific cpSSR markers that could help for studies on population genetics of the Fritillaria species.

Repeat sequences play an important role in genome rearrangement and variation due to the illegitimate recombination and slipped-strand mispairing in the chloroplast genome (Cavalier-Smith, 2002; Lu, Li & Qiu, 2016; Yuan et al., 2017). In the present repeat analysis, five types of repeats including tandem, palindrome, forward, reverse, and complement repeats were identified (Fig. 4). Among them, tandem repeats had the largest numbers and were mainly distributed in the intergenic spacer (IGS). Although substantial repeats have been distinguished in the chloroplast genome of higher plants, the mechanism for the origin of these tandem repeats was unclear (Vieira et al., 2014; Yi et al., 2013). Significant correlations have been observed among DNA rearrangement, mutation, and gene duplication (Cosner et al., 1997; Do, Kim & Kim, 2014; Vieira et al., 2014; Yi et al., 2013). It was reported that repeat sequences made sense for population genetics because of their significance in rearrangement (Cavalier-Smith, 2002). Most of the remaining four repeats occurred in intergenic spacer regions and the lengths ranged from 9 to 95 (4D–4F, Tables S5–S6). The results for the locations and sequence lengths of the four major repeats were similar to the latest studies (Bi et al., 2018; Park et al., 2017). The research also revealed that repeat sequences were caused by illegitimate recombination and slipped-strand mispairing in the genome (Cavalier-Smith, 2002; Lu, Li & Qiu, 2016; Yuan et al., 2017). Furthermore, the region where the repeats existed was a potential hotspot for genomic reconfiguration (Gao et al., 2009). Additionally, these repeat motifs might provide some informative sources to develop genetic markers for analysis on population genetics (Nie et al., 2012).

Expansion and contraction at the boundaries on the IR regions of the chloroplast genome are important factors that cause size variations and this plays a major role in structural stability and evolution (Asaf et al., 2018; Dang et al., 2014; Wang et al., 2008). In this study, a detailed comparison of four junctions of two IRs between F. cirrhosa and its closely related species was performed. The IR regions are highly conserved and structure variation was not significant in the eight Fritillaria chloroplast genomes (Fig. 6).

Identification of highly variable regions

Highly variable regions of the chloroplast genomes could not only be used for resolving phylogeny and identifying species at the species level, but also provide crucial information to explore species divergence and population structure at the population level (Dang et al., 2014; Du et al., 2017). Nucleotide diversity was calculated with a sliding window to estimate the divergence of different regions in eight Fritillaria cp genomes. Of these regions, the SSC region exhibited the highest value (0.00332) and the IR regions had the least (0.00038) (Table 5). Once again it indicated that IR regions were conserved in eight Fritillaria cp genomes. Similar results related to these regions have been reported in the latest studies of Fritillaria (Bi et al., 2018; Park et al., 2017) and have also been found in Lilium (Du et al., 2017). Furthermore, six relatively highly variable regions, including 5 intergenic regions (trnS-GCU-trnR-UCU, rpoB-psbD, rps4-trnF-GAA, petA-psbL, and ndhF-ndhD) and one gene region (ycf1) from the chloroplast genomes, were selected as potentially suitable gene fragments to study species identification and phylogenetics in Fritillaria (Fig. 8). The region of petA-psbL possessed the highest variability, followed by rps4-trnF-GAA, ndhF-ndhD, ycf1 and rpoB-psbD, whereas trnS-GCU-trnR-UCU was the lowest. Therefore, the regions with rich variation and suitable length, such as petA-psbL, rps4-trnF-GAA, ndhF-ndhD, ycf1 and rpoB-psbD could be used as a prior choice of species identification for Fritillaria. Meanwhile, all of the highly variable regions are judged to be suitable for revealing phylogenetic relationships and genetic structure at the species and population level in Fritillaria.

Phylogenetic analysis

In the present study, the four datasets from the plastid genomes, unanimously clustered Fritillaria cirrhosa and its closely related species into three clades (clade I, II and III) based on BI, MP, and ML analysis (Fig. 7). First, clade I was composed of two parts: F. sichuanica and F. dajinensis, as well as F. unibracteata and its variety. It was surprising to find that F. sichuanica was so closely related to F. dajinensis because they were obviously different in flower traits (Fig. 1) (Chen & Helen, 2000). One possible reason might be the conflict between molecules and morphology that was also observed in other taxa (Anand et al., 2016). The two species that were located at the northeastern edge of geographical distribution of the whole group, namely F. taipaiensis and F. yuzhongensis were clustered into clade II. Finally, F. cirrhosa was the most closely related to F. unibracteata and F. sinica. Although F. sichunica is thought to be a hybrid among F. cirrhosa, F. unibracteata and F. przewalskii (Luo & Chen, 1996), they seemly did not show close relationships. Our results were highly supportive of those from other studies and agreed with Huang et al. (2018) in the phylogeny of Fritillaria at the species level. However, among F. sichunica and its relatives, the phylogenetic inference of these results remained ambiguous. Molecular data from the nuclear genome and genetic analysis on population level might be necessary to further explore phylogenetic relationships among these related species.

Moreover, this study preliminarily indicated that the concept of the “complex group of F. chirrhosa” suggested by Luo & Chen (1996) might not contain four species but include other species as well (Fig. 8). Fritillaria cirrhosa is widely distributed in the alpine and subalpine regions of SW China, and exhibits complicated variations in morphology among the different distributions. Luo & Chen (1996) proposed the concept of a “complex group of Fritillaria cirrhosa”, including four species, namely F. cirrhosa, F. sichuanica, F. taipaiensis, and F. yuzhongensis based on obscure morphological traits and rough geographical distributions. However, there are no obvious borderlines among species within the complex group, as well as between the group and their closely related species. For example, F. sichunica is extremely similar to F. unibracteata except for subtle differences in length of the stigma lobes (Chen & Helen, 2000). So, the concept might be unreasonable and should be revised based on more detailed research.

Super and specific DNA barcodes

Potential DNA barcodes are generally used in species identification and phylogenetic studies of plants, but they could not provide enough informative sites to resolve the relationships among F. cirrhosa and its closely related species (Burke et al., 2016; Percy et al., 2014; Zhang et al., 2016). In recent research, the complete chloroplast genome as a super-barcode has been proven to be an effective tool for species discrimination in some complicated groups, and specific DNA barcodes are a trade-off for species identification of those groups based on highly variable regions of the plastid genome (Chen et al., 2018; Ma et al., 2018). In the genus Fritillaria, the complete chloroplast genomes were much better at uncovering the phylogeny of Fritillaria species (Bi et al., 2018; Li et al., 2018; Park et al., 2017). Similarly, clear phylogenetic relationships among F. cirrhosa and its close relatives were indicated based on these tools with extremely high bootstrap values in this study (Fig. 8). Thus, using the whole chloroplast genome as a super-barcode might be suitable for the species identification of Fritillaria. Meanwhile, highly variable regions observed in this study could be also used as specific barcodes for identifying species in Fritillaria.

Conclusion

The chloroplast genomes of F. cirrhosa and its closely related species were sequenced using NGS technology and their genetic information was primarily revealed. The eight genomes exhibited a typical circular quadripartite structure and shared a high similarity in gene order and genomic structure, but still provided rich genetic information for research on the Fritillaria species. The position change of the IR/SC junction was not obvious among the eight cp genomes. SSRs, large repeat sequences, and pairwise sequence divergences were determined. Highly variable loci and divergent regions were identified as possible ways to develop genetic markers which could be used for further study on population genetics. Moreover, phylogenetic analyses revealed that the eight Fritillaria species were divided into three clades with high support values based on the genome-scale datasets. The results indicated that F. cirrhosa was the close relative to F. unibracteata and F. sinica; thus, it indicated that the concept of a “complex group of F. chirrhosa” might be inappropriate and need further revision. Furthermore, the complete chloroplast genomes and highly variable regions were very promising for identifying the species and resolving phylogeny in F. cirrhosa which meant that they could be used as super-barcode and specific barcodes of the genus. Overall, the study would be beneficial to facilitate our understanding on phylogeny and evolution in Fritillaria.

Supplemental Information

Figure S1 Phylogenetic relationship of IR region of nine Fritillaria species

(A) Bayesian analysis (BI), (B) maximum parsimony (MP), and (C) maximum likelihood (ML).

Click here for additional data file.

Supplemental Information 1 The chloroplast whole genomes of Fritillaria spp

DNA sequences for the nine chloroplast whole genomes of Fritillaria spp. in this study (MH244906 –MH244913).

Click here for additional data file.

Table S1 Collection information of nine Fritillaria species

Click here for additional data file.

Table S2 Amino acid frequencies in protein coding genes of eight Fritillaria cp genomes

Click here for additional data file.

Table S3 Distribution of simple sequence repeats (SSRs) loci in the eight Fritillaria chloroplast genomes

Click here for additional data file.

Table S4 Regions of tandem repeat in eight Fritillaria chloroplast genomes

Click here for additional data file.

Table S5 A list of repeated sequences and their locations identified in the eight Fritillaria chloroplast genomes

Click here for additional data file.

Table S6 Frequency of complement repeats by length in eight Fritillaria

Click here for additional data file.

Table S7 Regions of highly variable sequences of Fritillaria

Click here for additional data file.

Table S8 Regions of highly variable sequences of Fritillaria

Click here for additional data file.

We thank Junbo Yang, Tingshuang Yi, Rong Zhang and Zhirong Zhang in Kunming Institute of Botany (Chinese Academy of Sciences, CAS) for their help in molecular experiment and data analysis of complete chloroplast genome in this study.

Additional Information and Declarations

Competing Interests

Author Contributions

DNA Deposition

Data Availability

The authors declare there are no competing interests.

Qi Chen performed the experiments, analyzed the data, contributed reagents/materials/analysis tools, prepared figures and/or tables.

Xiaobo Wu performed the experiments.

Dequan Zhang conceived and designed the experiments, contributed reagents/materials/analysis tools, authored or reviewed drafts of the paper, approved the final draft.

The following information was supplied regarding the deposition of DNA sequences:

All the sequences are accessible at GenBank via accession numbers MH244906, MH244907, MH244908, MH244909, MH244910, MH244911, MH244912 and MH244913. The sequencing data are also available as a Supplemental File.

The following information was supplied regarding data availability:

Specimens are available at the Herbarium of Medicinal Plants and Crude Drugs of the College of Pharmacy and Chemistry, Dali University under accession numbers ZDQ130053, ZDQ15022, ZDQ130018, ZDQ13030, HCB1, ZDQ14003, ZDQ15023, ZDQ15021, and ZDQ15009.

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
