# Peer review of "Phylogenetic analysis of Fritillaria cirrhosa D. Don and its closely related species based on complete chloroplast genomes"

_PeerJ, doi:10.7717/peerj.7480_

## Round 0.1 · original submission · Major Revisions

As you can see both referees see value in the work, but both also have rather extensive suggestions for improvement of the manuscript. In particular the scholarship of comparisons with previous research, the availability of data in public databases such as NCBI, the inclusion of single individuals of each species, and the conclusions regarding not supporting monophyly of the Fritillaria are all questioned by the referees. Such issues prevent publication of the manuscript in its current form, and it will require a major revision prior to additional review to address these issues.

[]

Reviewer 1 ·

Basic reporting

Genus and species names should be italicized.

1.There are very many stylistic issues in the references and citaion in manuscript.
2. In manuscript, 'chloroplast, plastome, plastid' words are uncertain. Authors must revise change.
3. Line 222, plstid genomes OR cp genomes... NOT whole plastid genomes....
4. Line 283 Single copy NOT SC
5. Authors should be check for Peerj format.

Experimental design

pass

Validity of the findings

pass

Additional comments

The manuscript by Qi and colleagues reports on the determination of chloroplast genomes from Fritillaria species with detailed analysis of phylogenetic relationships with other Fritillaria of this genus. Overall the manuscript is clearly written, although there are a few parts of the text that require modification.

Q1: Line 291-300, Expansion and contraction of other Fritillaria chloroplast genomes are indicated that difference results are compared to this paper. Authors should be check previous paper.

Q2: In section “Phylogenetic analysis”, more than half of them can’t be searched from NCBI. Especially, the species, F. cirrhosa, which the complete chloroplast genomes have been submitted to the NCBI, but they were not considered in this paper.

Q3: Line 441-451, authors mentioned about 'complex group of F. chirrhosa'. Recently, three papers for Fritillaria chloroplast genome analysis reported phylogenic analysis using chloroplast genoems. However, this results have weak informatiom. Also, complex group should be show repetition for same specise samples. This paper is one sample used for one speceis. Thus, discussions are carefuly consider.

Line 301-305, this paragraphe must change below to line 308.
Line 313 petA-psbL, check the size of letters
Line 323 Full name should be define before using abbreviation.
Line 395 - 397 It should be remove that sentence metioned in material and methods.
Line 413 - 423 This paragraphe should be move to part of Introduction.

Reviewer 2 ·

Basic reporting

There are many incorrect expressions in the manuscript, for example Line 69, line 100, line 286, line 362, line389-400, line 414 and etc. We strongly suggest the author to revised throughly. Second, the latin name should alway be italic type, however "var." should be upright type. There are several places need to be revised, such as line 98, line 105, line 191, line 215 and etc. Furthermore there are other nonstandard place that marked in the pdf file need to be carelly revised.

Experimental design

One of the major conclusion in the manuscript is "this study did not support the so-called “complex group of Fritillaria cirrhosa”, 442 suggested by Luo & Chen (1996a) as a monophyletic group", according to Fig.8, we bellieve that it lack abundant proof.

Validity of the findings

The findings in this manuscript provide a new method to to accurately identify F. cirrhosa and its closely related species.However, in our opinion, some of the closely related species are also traditional Chinese medicine.The authors need to state clearly what is the new findings which can expand our knowledge about F. cirrhosa

Additional comments

The findings has significance in identifying F. cirrhosa and its closely related species, however, we believe it lacks novelty. There are many mistakes and noncosistence in the writing which need significant revisions. And the gene number in Line 222 is not accurate as there are reduplicated genes which should be excluded.

Annotated reviews are not available for download in order to protect the identity of reviewers who chose to remain anonymous.

---

## Round 0.2 · accepted · Accept

As you can see both referees are satisfied with your revisions and now recommend that the paper be accepted. There remain some minor edits for typos, tense, font, etc., but I believe these can be handled in the proofing stage, so I am happy to move your manuscript forward into production. Congratulations, and thanks for submitting your work to PeerJ.

Reviewer 1 ·

Basic reporting

This paper is well changed than 1st paper.

Experimental design

no comment

Validity of the findings

no comment

Additional comments

no comment

Reviewer 2 ·

Basic reporting

1.The title in word version is not consistent with that in PDF version.

Experimental design

'no comment'

Validity of the findings

'no comment'

Additional comments

line 251, rRAN or rRMA?
lin 313-314 'the single copy regions' Font sizes are inconsistent.